# Study design and baseline to evaluate water service provision among peri-urban communities in Kasai Oriental, Democratic Republic of the Congo

Kathleen Kirsch[1]*, Corey Nagel[3], Chantal Iribagiza[1], John Ecklu[1], Ghislaine Akonkwa Zawadi[1], Pacifique Mugaruka Ntabaza[1], Christina Barstow[1], Andrea J. Lund[2], James Harper[1], Elizabeth Carlton[2], Amy Javernick-Will[1], Karl Linden[1], Evan Thomas[1]*

1 Mortenson Center in Global Engineering and Resilience, University of Colorado Boulder, Boulder, Colorado, United States of America, 2 Environmental and Occupational Health, Colorado School of Public Health, Denver, Colorado, United States of America, 3 Department of Biostatistics, College of Public Health, University of Arkansas for Medical Sciences, Little Rock, Arkansas, United States of America

* ethomas@colorado.edu (ET); Kathleen.Kirsch@colorado.edu (KK)

**Data Availability Statement:** Data cannot be shared publicly because of human subjects restrictions. Data are available from the University

## Abstract

We present a study design and baseline results to establish the impact of interventions on peri-urban water access, security and quality in Kasai Oriental province of the Democratic Republic of the Congo. In standard development practice, program performance is tracked via monitoring and evaluation frameworks of varying sophistication and rigor. Monitoring and evaluation, while usually occurring nearly concurrently with program delivery, may or may not measure parameters that can identify performance with respect to the project's overall goals. Impact evaluations, often using tightly controlled trial designs and conducted over years, challenge iterative program evolution. This study will pilot an implementation science impact evaluation approach in the areas immediately surrounding 14 water service providers, at each surveying 100 randomly-selected households and conducting water quality assessments at 25 randomly-selected households and five water points every three months. We present preliminary point-of-collection and point-of-use baseline data. This study is utilizing a variety of short- and medium-term monitoring and impact evaluation methods to provide feedback at multiple points during the intervention. Rapid feedback monitoring will assess the continuity of water services, point-of-consumption and point-of-collection microbial water quality, household water security, household measures of health status, ability and willingness to pay for water and sanitation service provision, and service performance monitoring. Long-term evaluation will focus on the use of qualitative comparative analysis whereby we will investigate the combination of factors that lead to improved water access, security and quality.

of Colorado Boulder Institutional Review Board for researchers who meet the criteria for access to confidential data. Email: irbadmin@colorado.edu; Phone: 303.735.3702.

**Funding:** This work is funded by the United States Agency for International Development.

**Competing interests:** This study is funded by the U.S. Agency for International Development under the terms of contract Number: 72066020C00001 and implemented by Chemonics. The funders have no role in study design, randomization, data collection and analysis, or decision to publish.

## Introduction

Research has highlighted some of the barriers to downstream point-of-use benefits in water, sanitation and hygiene (WASH) interventions. Despite improvements at water service providers (WSPs), there can still be issues of intermittent water supply, contamination from collection to storage, differences in benefits across network types and sizes, and a need to incorporate community-led interventions across water, sanitation, and hygiene [1–3]. While work with water service providers may target the three T's of WASH finance (tariffs, taxes and transfers) it is important to consider household investment and its impact on whether households achieve access to adequate WASH levels [4]. Even with governance, financial, and infrastructure support, this household investment may be critical to see any point-of-use benefits.

Despite WASH policies and planning, national WASH sectors generally lack the funding for implementation. When assessing national governance and financing, the United Nations and World Health Organization found less than 15% of 115 surveyed countries had the needed WASH resources, some reporting funding gaps of up to 60% [5]. Despite ambitious targets for universal access to water and sanitation by 2030, meeting these targets would involve an estimated $114 billion each year of capital investments, or about three times the amount of current investment levels [6].

As public and private organizations look to design projects to support service delivery, their efforts are hampered by long lag times in research. Randomized controlled trials, for example, can be prohibitively expensive for programs and require additional rigidity during implementation. There is a need for implementation science which can achieve causal inference while allowing the required targeted interventions flexibility during implementation.

The U.S. Agency for International Development (USAID) launched the Accelerating Peri-Urban Water and Sanitation Services in Kasai Oriental and Lomami Provinces Activity. From 2020–2025, this program will work with WSPs and relevant municipal and provincial governments in Kasai Oriental and Lomami provinces in the Democratic Republic of the Congo (DRC) to improve their water service delivery and performance using a range of targeted interventions. The University of Colorado Boulder is providing the implementer, Chemonics, with research support to introduce innovations for evaluating service provision in real-time, conduct rapid impact evaluations of these activities to inform second-stage interventions within the program and improve water access, use and quality. Assessed interventions may include support for WSP financial capacity, institutional governance strengthening, and water and sanitation infrastructure and equipment support [7].

The DRC has 50% of the African continent's water reserves, but only 52% of the population has access to an improved water source [8]. These challenges are currently exacerbated by the existing COVID-19 pandemic and a cholera outbreak. Lack of access to WASH is among the top five risks associated with death and disability in the DRC, and this is further aggravated by the inherent challenges of peri-urban service provision in the provinces of Kasai Oriental and Lomami [9]. The DRC's peri-urban areas exist in a gap between the parastatal company Régie de Distribution d'Eau's (REGIDESO) mandate for urban water provision and other governmental and non-governmental programs for rural water services. Lomami and Kasai Oriental Provinces face urbanizing rural areas coupled with expanding peri-urban areas, and approximately 2.5 million people lack access to water service provision from either REGIDESO or local water management committees. These households rely on informal systems of basic services [10]. A recent baseline assessment of 769 water service providers in these provinces found that only 4.6% reported any infrastructure beyond a handpump, and that there was a need for technical assistance in administration, governance and financial management [10].

The University of Colorado Boulder will utilize a number of methodologies to conduct real-time, rapid impact evaluations of these activities to inform second-stage programmatic interventions. These methods will leverage statistical approaches such as interrupted time series (ITS). There is increasing use of ITS to evaluate large-scale national policies and programs without the presence of a control group. The ITS outcome variable is measured repeatedly pre- and post-intervention, incorporating the prior observational study parameters and analyzing slope changes before and after to see how the outcome variable behavior is altered. After a statistically significant observation period, additional intervention events can be introduced [11]. In the DRC, a recent ITS evaluation found a free care policy introduced by the Ministry of Public Health during the 2018 Ebola outbreak was effective at quickly increasing the use of certain health services, though this trend was not sustained after the program's end [12]. Another ITS evaluation in South Africa assessing a natural experiment of on-site access to pre-exposure prophylaxis (PrEP), found this was associated with halving HIV incidence, though the study acknowledged that access to PrEP was confounded with calendar time so the possibility of falling HIV incidence as partly explained as a cohort effect could not be entirely ruled out [13]. Trying to minimize confounders and extend ITS to causal inference remains a challenge.

Our study seeks to assess the effectiveness of interventions to improve water access, use and quality. Planned intervention activities are specific to the existing capacity of each WSP. These may include support to improve infrastructure, financial capacity, institutional governance, and other assistance as needed. To assess these interventions, this research utilizes short- and medium-term monitoring and impact evaluation methods in order to provide feedback at multiple points during the intervention. Short-term methods provide rapid feedback to improve the intervention during the implementation cycle, where medium-term evaluation will assess the intervention from a systems-level perspective.

The rapid feedback monitoring will produce periodic evidence-based reports that demonstrate the outcomes and impacts of water and sanitation delivery activities, including continuity of water services, point-of-consumption and point-of-collection microbial water quality, household water security, measures for household health status, ability and willingness to pay for water service provision, and WSP performance monitoring. Medium-term evaluation focuses on the use of qualitative comparative analysis (QCA) whereby the combination of factors that lead to successful or unsuccessful program outcomes can be posited.

We present a study design to establish the impact of interventions on peri-urban water service provision, as well as preliminary point-of-collection and point-of-use baseline data. Ultimately, this study seeks to pilot and assess an implementation science impact evaluation approach in the areas immediately surrounding 14 water service providers, utilizing a variety of short-, medium-, and long-term monitoring and impact evaluation methods to provide feedback at multiple points during the intervention. This research will further assess to what degree these feedback mechanisms affect future intervention effectiveness with WSPs, such as SADEL WSP (Fig 1). The study may not have a sufficient adjustment set to draw all the causal conclusions, but this research will use the best available data and analyses possible while allowing for flexibility in confounding variables. Ultimately, this study seeks to provide rapid performance information to inform next-stage interventions within the existing activity, providing foundational evidence to support development actors in future programs on not just the effectiveness of these water service provision interventions, but on how these rigorous analyses can be executed to inform decisions during implementation. The methodology presented in the next section, and its subsequent results, seeks to augment existing scientific literature and improve WSPs' performance in pursuit of Sustainable Development Goal 6, ensuring availability and sustainable management of water and sanitation for all [14].

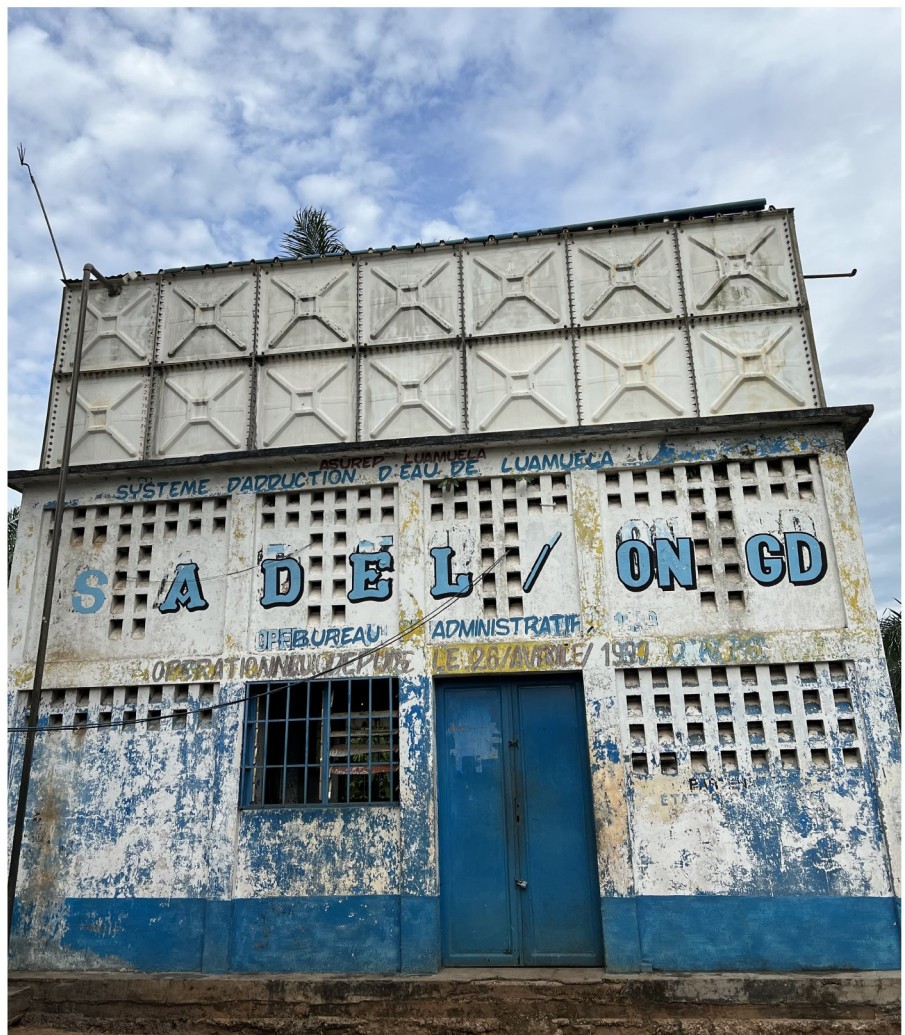

**Fig 1. SADEL water service provider.** SADEL, one of the 32 water service providers included in the Accelerating Peri-Urban Water and Sanitation Services in Kasai Oriental and Lomami Provinces Activity. This water service provider is included in the cohort of 14 providers in the University of Colorado Boulder study and is located in Kasai Oriental Province.

## Materials and methods

### Study setting

Located in the Kasai Oriental province of the Democratic Republic of the Congo, this study assesses interventions with water service providers in peri-urban areas. A baseline assessment was conducted by Chemonics during inception workshops held in both provinces in September 2020. Their assessment included WSP mapping, capacity assessments, household water consumption surveys, and a financial intermediary landscape analysis. The assessment identified 769 WSPs: 418 in Kasai Oriental and 351 in Lomami. Most of the infrastructure for these WSPs were hand pumps. Ultimately, 32 WSPs were selected by Chemonics for inclusion in the intervention program based on subjective programmatic criteria including need and implementation feasibility. Due to the complex logistics of implementing this program across multiple sites in the region, initiation and delivery of program activities will be staggered over a

period of 4 years (2021–2025), with successive tranches of WSPs beginning program activities at roughly three- to six-month intervals. Neither the selection of WSPs for inclusion in the intervention program nor the timing of program delivery can be randomized given existing logistical and political conditions in the region.

## Study design

We have selected a non-randomized stepped-wedge study design as the approach that best aligns with the staggered schedule of program initiation required by the the program implementer [15]. Further, this study design will yield multiple rounds of both pre- and post-implementation data that can be used in a variety of analytic approaches and is flexible to potential changes/modifications in the timing of program activities [16].

The 14 WSPs to be included in the current study are WSPs receiving program activities in Kasai Oriental. This will allow the study adequate time to collect pre- and post-intervention data. Specifically, stepped-wedge design contains four steps, each step consisting of a group of three WSPs, with data collection in the included WSPs occurring at three-month intervals, for a total of 6 rounds of data collection (Fig 2).

Chemonics conducted a WSP service area census, which included capturing the location and general information for every household located within 500 meters of standpipes operated by the WSPs. For the first cluster of WSPs, this census was used as population-level data to pull the baseline sample from. Households were randomly selected, and the sampling target was set at approximately 200 households per WSP. Fig 3 shows the households surveyed in the ACAEL service area, with the 500 meter radius from the standpipes shaded.

## Sample size

Within each WSP, we are collecting data at three-month intervals on a cohort of 100 randomly selected households for the duration of the study period within 500m of the WSP. The households randomly selected at baseline are retained over the course of the study. Using an open-cohort design, if a household is unavailable after three visits, another household is randomly selected. This three-month assessment includes all tools such as the point-of-collection survey, the household survey, and water quality assessments. Water quality testing is performed in a randomly selected sub-sample of 25 households per WSP.

In order to maintain adequate sample size, we adopted an open-cohort approach where we replace households that are lost to follow-up with newly selected households when necessary [17]. This is particularly important because we observed a high rate of household migration in and out of some study areas, for example due to reliance on location-shifting mining activities.

We determined our required sample size (both the number of WSPs and the number of households) based on three primary outcomes: 1) the proportion of household water samples with thermotolerant levels at or greater than the WHO high risk category, 2) the proportion of households meeting the threshold for water insecurity, and 3) the proportion of children under five years of age with diarrhea in the past seven days. Although we employ an open-cohort design, we calculated the sample size required for a repeated cross-sectional design as this is a more conservative approach and, we believe, appropriate in the absence of reliable projections of attrition rates across the study period [18, 19]. Calculations were performed using formula published by Hemmings et al and implemented in the Shiny CRT calculator in R [20]. We assumed a block exchangeable autocorrelation structure with an intracluster correlation coefficient (ICC) of.10 and a cluster autocorrelation coefficient (CAC) of 0.70 for water quality and water insecurity, and an ICC of 0.05 and a CAC of 0.70 for diarrhea, an alpha of

| Group # | # | WSP Name | Schedule | | | | | | | | | |
|---|---|---|---|---|---|---|---|---|---|---|---|---|
| | | | Q4-21 | Q2-22 | Q3-22 | Q4-22 | Q1-23 | Q2-23 | Q3-23 | Q4-23 | Q1-24 | Q2-24 |
| 1 | 1 | ACAEL | 0 | 0 | - | 0 | 0 | 1 | 1 | 1 | 1 | 1 |
| | 2 | SADEL | 0 | 0 | - | 0 | 0 | 1 | 1 | 1 | 1 | 1 |
| 2 | 3 | CAEPT | - | 0 | - | - | 0 | 0 | 1 | 1 | 1 | 1 |
| | 4 | FOMI-Mbuji Mayi | - | 0 | - | - | 0 | 0 | 1 | 1 | 1 | 1 |
| | 5 | Jamaique & Kanda Kanda | - | - | - | - | 0 | 0 | 1 | 1 | 1 | 1 |
| | 6 | Passoire | - | - | - | - | 0 | 0 | 0 | 1 | 1 | 1 |
| | 7 | Musololu | - | - | - | - | 0 | 0 | 0 | 1 | 1 | 1 |
| | 8 | Forage 100 Jours | - | - | - | - | 0 | 0 | 0 | 1 | 1 | 1 |
| 3 | 9 | Kabue | - | - | - | - | 0 | 0 | 0 | 0 | 1 | 1 |
| | 10 | ACODEL | - | - | - | - | 0 | 0 | 0 | 0 | 1 | 1 |
| | 11 | KASULU | - | - | - | - | 0 | 0 | 0 | 0 | 1 | 1 |
| | 12 | Paroisse Marc | - | - | - | - | 0 | 0 | 0 | 0 | 0 | 1 |
| | 13 | Tshitandayi | - | - | - | - | 0 | 0 | 0 | 0 | 0 | 1 |
| | 14 | FOMI-Tshitenge | - | 0 | - | 0 | 0 | 0 | 0 | 0 | 0 | 1 |

**Fig 2. Data collection schedule.** Data collection schedule for the 14 WSPs included in the University of Colorado Boulder study design from 2022–2025 surveying rounds.

0.05 and a power of 0.8. We derived ICC and CAC estimates from our baseline data in the region and previous work in similar settings.

Given the stepped wedge design previously specified and the parameters presented above, we are powered to detect absolute reductions of 7.3 percentage points in household water insecurity (from 90.0% to 82.7%), 10.5 percentage points in household water quality samples at or above the WHO high risk threshold (from 87.0% to 76.5%), and 7.6 percentage points in diarrhea (20.0% to 12.4%). However, we emphasize that these estimates are based on the very conservative assumption of repeated cross-sections at each observation period, and so may underestimate the observed power that will result from the planned open-cohort design.

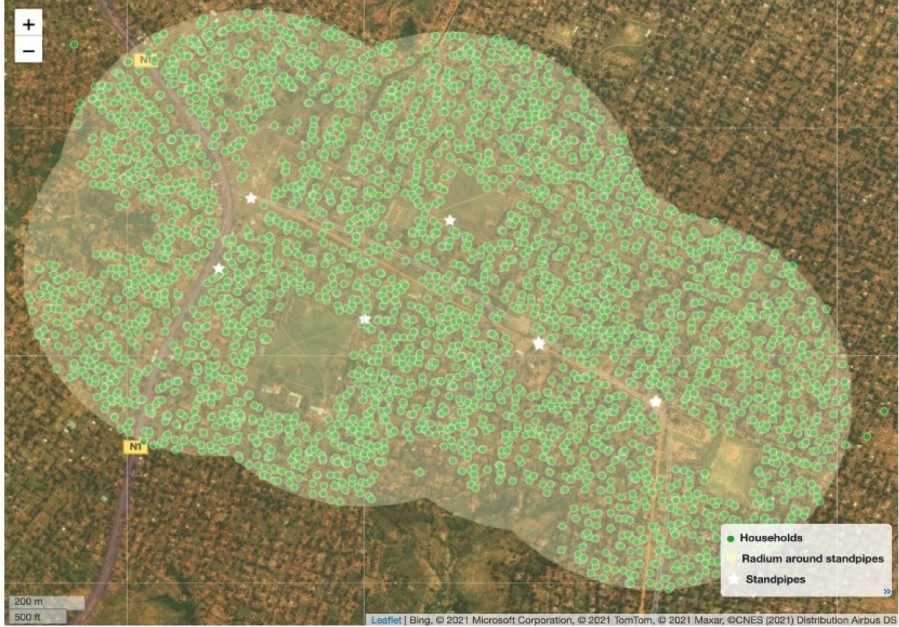

**Fig 3. Household surveying near ACAEL water service provider.** Households surveyed during census within 500 meters of ACAEL water service provider standpipes in Kasai Oriental Province.

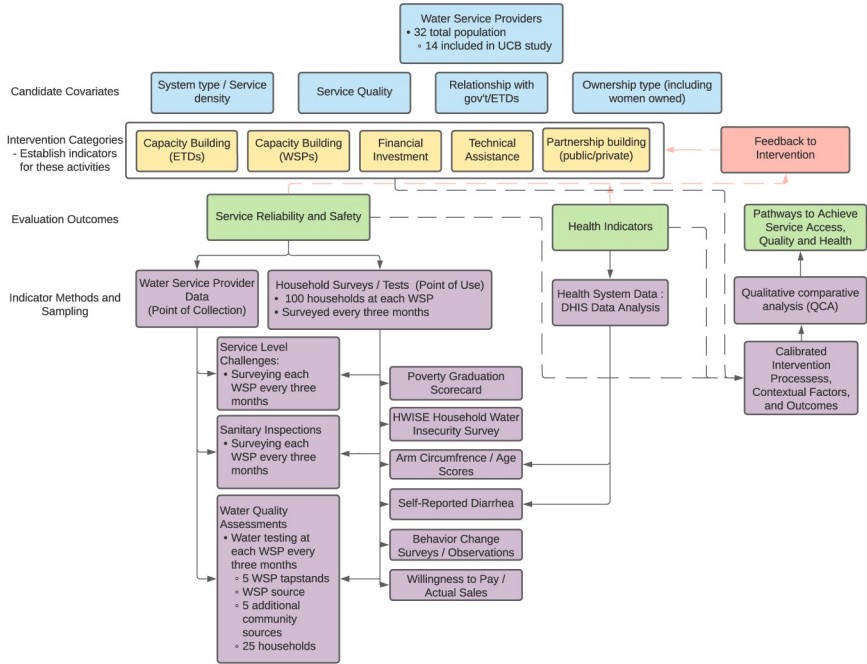

**Fig 4. Research flowchart.** Flowchart of research covariates, interventions, metrics, and primary evaluation outputs for the study design of the 14 WSPs in Kasai Oriental province.

## Study metrics

Fig 4 below provides a holistic view of the covariate characteristics, interventions, primary evaluation outputs, and specific metrics which influence evaluation outputs. All survey data is collected using mWater and response data is pulled directly into R for analysis.

Primary research outcomes include:

- Assessing whether the interventions with WSPs result in improved water access, use and quality.

- Investigating whether improvements in point-of-collection water access, use and quality result in improved point-of-use water access and quality.

- Evaluating the ability to estimate causal effects using the observational data collected in the proposed study design.

## Study components

Point-of-collection surveys are conducted at the 14 water service providers and point-of-use surveys are conducted at the selected households every three months. The point-of-collection surveys target nearby users currently collecting water from the water service provider. The following sections describe some of the tools built into these surveys.

**Poverty scorecard tool.** The Scorocs Simple Poverty Scorecard tool is validated and is used in the household survey to estimate and track poverty rates over time at the household level [21, 22]. It was specifically created for the Democratic Republic of Congo and uses 10 indicators to estimate the likelihood that a household has consumption below a given poverty line [23]. Both self-reported questions and observations are used to determine the poverty

likelihood as well as the international poverty line of a specific household. This tool is implemented every three months.

**Household Water Insecurity Experiences Scale.** The Household Water Insecurity Experiences (HWISE) scale is used to assess the magnitude of water insecurity within communities, track their changes over time, and inform implementation and policy development. The 12 questions target information such as how water availability affects hygiene, planning, stress, etc. and which are summed to develop a household's scale score. Higher scores indicate greater water insecurity, and HWISE developed a provisional cut-point of 12 or higher for water-insecure households based upon prior work assessing water situation satisfaction, perceived stress, and food insecurity [24]. HWISE is assessed every three months across all study households.

Household water insecurity has been associated with adverse consequences related to both physical and mental health. Stressors caused by household water insecurity have been shown to undermine human health and development by influencing common mental disorders, and throughout key maternal and infant health stages from pregnancy to infant feeding practices [25–30]. Thus, measurement of household water insecurity is an important holistic tool to assess of a household's well-being and inform water-related interventions. HWISE has been validated at 29 sites across 23 low- and middle-income countries and thus creates a comparable tool to use in a diversity of environments [24]. This is included in regular monitoring as a key metric to track intervention progress and effectiveness.

**Water, sanitation and hygiene practices.** Sanitary inspections occur at both points-of-use and points-of-consumption according to the newly revised World Health Organization small water supplier sanitary inspection and household inspection best practices [31, 32]. Risks are identified across questions assessing water collection, storage, treatment and handling. Greater scores correlate to increased risk. Additionally, the survey asks individuals questions to assess their perceptions of using clean water, such as risk or importance of clean water, confidence in availability, and planning for clean water collection.

At the water service provider and household level this research study assesses aspects of quantity, quality, availability, accessibility, and behavioral practices. The evaluation of the water service will be driven by the Sustainable Development Goal (SDG) framework, specifically Target 6.1, "By 2030, achieve universal and equitable access to safe and affordable drinking water for all" [33]. To benchmark and compare services, the Joint Monitoring Program (JMP) service ladders will be utilized based on its four classification criteria: 1) improved drinking water source (such as piped water, boreholes, protected dug wells or springs, rainwater, etc.), 2) accessibility of water source, 3) availability of water, and 4) quality of water [34]. WASH practice information is collected every three months.

**Willingness to pay.** For WSPs or any other market-based program, the decision to purchase a good or service is of interest and is typically based on different factors, as described by behavioral science [35]. Whether an individual has the required opportunities and resources to purchase a specific good or service, or their ability to pay, is a factor of "actual control" [36]. This factor is determined by the available financial resources of an individual in the context of the various other conflicting purchases made in daily life. An individual's ability to pay can be measured by calculating their income, assets, and expenses and incorporating any expected changes [37].

Conversely, willingness to pay (WTP) is considered a factor of "perceived control", or an individual's perceived ease or difficulty of performing a certain behavior [36]. Perceived control is affected by characteristics including the good or service's cost, quality, affordability, and the individual's ability to pay [37]. While ability to pay affects WTP, evaluating WTP is critical for WSPs. Stated WTP (what people say they will pay for a good or service) can be investigated using a variety of methods, but the most indicative of revealed WTP (what people actually pay

for a good or service) is widely considered to be a discrete choice experiment (DCE) [38]. In DCE questions, options are described by attributes with different levels, and consumers are asked to choose preferred options. In this DCE field evaluation, WSP consumers are asked to choose options with varying availability (always vs sometimes), potability (yes vs no) and pricing (0 to 400 CDF). The DCE evaluation is conducted in selected households in the areas surrounding each WSP every three months.

**Health.** Health metrics are gathered in parallel with the implementation water service delivery interventions. Improved access to safe water and sanitation can reduce exposure to enteric pathogens leading to reductions in diarrhea [39] and malnutrition [40], which are major drivers of mortality in children under age five [41]. The realized health impacts of improved water and sanitation vary by pathogen and key exposure pathways [42], and recent evidence suggests that even well-adopted WASH interventions can fail to meaningfully alter child health outcomes in areas where environmental exposure to fecal pathogens is high [43].

Mid-Upper Arm Circumference (MUAC) is the circumference of the upper arm, measured at the mid-point between the tip of the shoulder and the tip of the elbow. MUAC is used for the assessment of nutritional status for children between the ages of six months and five years old. Additional questions are asked on recent health parameters such as fever and diarrhea. These assessments focus on recent diarrheal events due to the imprecision of long-term recall. Household-level assessments of diarrheal illness are complemented by community-level public health records if data is available. All health metrics are collected at households every three months.

**Water quality assessments.** Water quality assessments are conducted at both household points-of-consumption and water service provider points-of-collection. These temporal household water quality assessments will target several parameters including turbidity, pH, conductivity, free and total chlorine, nitrate, nitrite, manganese, arsenic, fluoride, and thermo-tolerant coliforms. We test bacteriological parameters in triplicate and chemical parameters in duplicate. Water quality assessments help locate possible areas of contamination and inform possible interventions on treatment and storage practices.

To align with WHO and UNICEF Joint Monitoring Programme guidance for "safely managed drinking water service," this designation requires:

"Households using improved drinking water sources which are located on premises, with water available when needed, and free from contamination*, are classified as having safely managed services. Households not meeting all of these criteria, but using an improved source with water collection times of no more than 30 minutes per round trip are classified as having basic services, and those using improved sources with water collection times exceeding 30 minutes are classified as limited services."

For water quality parameters, this contamination is defined as "the absence of faecal indicator bacteria (E. coli or thermotolerant coliforms), and data on arsenic and fluoride will also be used where available" [44].

The baseline WSP water quality assessment will ensure there are no arsenic or fluoride concerns, as well as testing for the absence of thermotolerant coliforms in a 100 mL sample. Water quality assessments are conducted on 25 randomly selected households per WSP from the household survey cohort every three months. In addition to these 25 households, water quality testing will be performed every three months on the WSP source, up to five WSP tapstands, and up to five additional community sources identified during surveying.

**Qualitative comparative analysis.** A fuzzy set Qualitative Comparative Analysis study (fsQCA) is used to analyze data on WSPs to determine which combinations of conditions the amount and type of funding received led to different outcomes sought in this work. This can include conditions such as capacity development interventions and financial reporting

mechanisms and types of funding such as OPX versus CAPX. Possible outcomes analyzed include access to water services (basic and safely managed), service quality, enhanced WSP capacity, etc. This effort will analyze individual conditions and outcomes for each of the studied WSPs, perform a cross-case comparison, and determine the pathways or 'recipes' of conditions that, in isolation or combined, associate with outcomes assessed near the end of the contract.

To conduct the analysis, we primarily rely upon data collected from Chemonics through other efforts, such as baseline and continued assessments of WSPs in areas of existing infrastructure and infrastructure management, financial management, administration, human resources management, and governance. This involves careful documentation and observation of capacity development interventions (such as developing and implementing a water safety plan and local contextual conditions) as well as assess and reflect sessions with the WSPs. In addition, outcomes are determined from other research partners engaged in the work, including water quality, quantity, and access. Each condition and outcome is then calibrated for each case. Importantly, calibration can be done for both qualitative and quantitative data. To do this, we define set anchor points for in-set membership (score of 1), out-of-set membership (value of 0) and the cross over point (score of 0.5). Quantitative data can also be calibrated by normalizing the data within anchor points. For instance, in a study in Uganda [45] on factors influencing revenue collection for preventive maintenance of community water systems, we calibrated the condition of "no alternative water sources available nearby" as follows: out-of-set membership (value of 0) was if there are nearby functioning improved water sources; partially out-of-set (value of 0.33) was if there are nearby water sources, but they are poor functioning or of poor water quality; and in-set (value of 1) was if there are no nearby improved water sources. After analysis, we found this factor to have the highest necessity score, as it was present in almost all cases that lead to successful compliance.

Thus, we use fuzzy-set logic based upon theoretical and practical cut off points to define set membership and assign values for each condition and outcome to each case (WSP) to build a 'truth table'. We then use the fsQCA software and case knowledge, relying on partner and field knowledge for interpretation as well, to minimize the truth table and to calculate, using Boolean algebra, pathways of combined conditions that lead to successful outcomes. Minimization would allow us to remove the least-important causal conditions and simplify pathways. We will then assess QCA metrics. For instance, necessity evaluates how commonly a causal condition is present with an outcome while coverage helps evaluate the generalizability of findings. Where cases have the same outcome, it is the fraction explained by the same pathway; where higher coverage indicates that the pathway explains more cases. Consistency evaluates each pathway's reliability and it is the fraction of cases that exhibit the same pathway and outcome. The result of these metrics is a 'recipe' of what pathways of combined conditions led to desired outcomes, or what individual or combined factors led to desired (and conversely non-desired) outcomes. We intend to discuss, interpret, and validate these findings via partners in the field.

## Ethics statement

This study was reviewed and approved by the University of Colorado Boulder Institutional Review Board and the Lomami and Kasai Oriental Provincial Health Division. As approved by the review boards, survey participants provide verbal informed consent during each survey and confirm they are at least 18 years of age and are knowledgeable about household water practices. Consent is marked in the mWater surveyor in enumerator tablets and data is managed in mWater.

This study protocol was approved by the University of Colorado Boulder Institutional Review Board (20–0491 approved May 11, 2021).

## Statistical methods

We will examine differences in the outcomes of interest before and after the implementation of the program using generalized linear mixed models. Given the open-cohort design, with repeated observations of households clustered within WSPs, all models will include random effects for both WSP and household, as well as fixed effects for step and season in order to account for secular and seasonal trends in the outcomes of interest [46]. We will estimate mixed effects models using adaptive Gaussian–Hermite approximation to the likelihood in order to account for the small number of clusters [47]. All models will be adjusted for both individual and WSP-level characteristics in order to minimize potential selection bias and confounding resulting from the non-random timing of program delivery. These include water source type, water quality and reliability as well as categorized management models. The primary analyses will be carried out following the intent to treat principle, with outcomes of interest evaluated during the baseline or evaluation periods irrespective of whether or not all program activities were implemented.

Different estimation strategies will be assessed. One possible parameter and estimation strategy pairing would be simple average treatment effect style parameters estimated using doubly robust estimation strategies (e.g. Targeted Minimum Loss Based Estimation, and Augmented Inverse Probability of Treatment Weighting), that seek to learn from both the intervention mechanism and the outcome mechanism. Another potential estimation strategy is interrupted-time series, estimating impacts on the outcome variable using pre- and post-intervention slope changes.

In addition to these primary analyses, we will conduct additional analyses examining both overall program impacts and those related to specific program activities across the range of outcomes presented. Our ability to conduct these additional analyses is facilitated by the structure and timing of data collection in a stepped wedge design, as it results in data that can be analyzed both horizontally (i.e. comparison of pre- and post-intervention periods/trends within clusters) and vertically (i.e. contemporaneous comparison of control and intervention clusters at specific time points). In formulating these additional analyses, we will use the causal inference road-map approach to define causal parameters of interest, determine appropriate statistical methods (e.g. inverse probability weighting, targeted maximum likelihood, interrupted time series, etc.) to estimate causal effects, and identify the conditions/assumptions required to draw causal conclusions from the resulting statistical estimates [48].

Further, we will incorporate geospatial and temporal characteristics including rainfall to establish any seasonality attributes associated with the outcomes measured.

## Results and discussion

The following preliminary baseline data was assessed from household surveys (n = 830), point-of-collection surveys (n = 70), and water quality assessments (n = 231) completed within the 500m area around five WSPs in Kasai Oriental.

### Demographics and poverty scorecard

The WSP areas had similar poverty scorecard results with an overall median score of 24 across all WSPs, representing 93.3% of the population below the $1.90 per day 2011 PPP "very poor" poverty line. The lowest poverty scorecard score was seen near CAEPT, where 94.8% of the surveyed population was experiencing the "very poor" definition, and the highest score was

seen near FOMI Mbuji Mayi at 79.1%. Across WSPs, the physical household is typically a packed earth/straw floor with mud brick walls. About half of the surveyed households have agricultural land (51%) and about 39% have livestock.

## Demographics and poverty scorecard

The HWISE assessment summarizes the number of households classified as water insecure in the areas surrounding each WSP. Table 1 indicates which households had a score of 12 or higher, the HWISE cut-point for a water insecure household. About 85% of surveyed households were classified as water insecure. In particular, ACAEL, CAEPT, and FOMI Tshitenge all had over 90% of surveyed households classified as water insecure. FOMI Mbuji Mayi had the lowest proportion of households with the HWISE water insecurity classification, at 37%.

## Water, sanitation, and hygiene practices

The average household size was seven people, with the primary water source at the time of surveying indicated to be a public tap/standpipe (36%) improved source, but there were notable unimproved sources used such as a river (32%) or unprotected spring (19%). When asked if they collected water from a WSP in the last two weeks, only 42% of respondents had. The areas with the greatest current usage of WSPs were near FOMI Mbuji Mayi (96%) and CAEPT (90%).

Households (83%) reported spending over 30 minutes to travel to their primary water source, queue, and then travel home, with 46% indicating it took 1–3 hours total. Given all households surveyed were within 500 meters of a WSP standpipe, either households are spending a large amount of time queuing at WSP standpipes or traveling large distances to use another primary water source. Further investigation is advised to determine if long wait times are possibly disincentivizing households from using WSP standpipes.

Approximately 37% households use an improved sanitation facility, such as a pour flush toilet or pit latrines with slabs. However, the majority (62%) of respondents share a toilet facility.

Sanitary inspections were conducted according to the newly revised World Health Organization household inspection best practices. Household risks are identified across questions assessing water collection, storage, treatment and handling. Greater scores correlate to increased risk. Most households did not use final or bulk storage containers. Scores were averaged across the number of questions answered with greater scores correlating to increased risk. The mean sanitary inspection score was 0.62 per household (SD 0.22), indicating risks were identified across the majority of questions during the inspection. The greatest risks were identified at sanitary inspection surveys of CAEPT (0.8) with similar median sanitary inspection scores at the other four WSPs (0.6). The three most common risks identified were collecting water from multiple sources, keeping the collection container in a place where it can become contaminated, and inadequately covering the collection container.

**Table 1. Number (%) of households classified as water insecure by HWISE and stratified by WSP area, n = 830.**

| WSP Area | n (%) |
|---|---|
| SADEL | 210 (88) |
| ACAEL | 202 (93) |
| FOMI Mbuji Mayi | 37 (37) |
| CAEPT | 97 (97) |
| FOMI Tshitenge | 160 (92) |

Using some WHO sanitary inspection indices for drinking water sources, enumerators conducted inspections at WSP points of collection. As shown in Table 2, enumerators identified heavy vegetation, unclean tap attachments, and signs of nearby pollution near collection points. In particular, inadequate drainage and inadequate fencing and barriers were identified in over 95% of inspections.

For the point of collection survey, enumerators conducted informal focus groups with water users at the collection points during the time of surveying. Around 37% of users indicated they did not believe the water they were collecting was safe. Water reliability fluctuated, as water was available on average 8 hours per day, and 77% of users reported seasonal functionality changes. A little over half (57%) of the collection points had meters.

Also identified during surveying, the cost of a jerry can of water varied considerably. Users reported a cost of 100 CDF (0.05 USD as of December 2022) to 300 CDF (0.15 USD as of December 2022). Approximately half of users reported using the collection point less during the rainy season.

## Willingness to pay

A preliminary conditional logit model of the discrete choice experiment found statistically significant preferences for all attributes (Table 3). Across all responses, households were 1.37±0.05 times more likely to prefer reliable water with a willingness to pay of 532±49 CDF (0.26±0.02 USD January 2023) for a 20L jerry can, and 1.39±0.05 times more likely to prefer purified water with a willingness to pay of 540±50 CDF (0.27±0.02 USD January 2023) for a 20L jerry can. Disaggregating by town, preferences and WTPs were found to range widely. Households in Luamuela had the strongest preference for reliable water but the weakest preference for purified water. Households in Lukalaba had the weakest preference for reliable water, while those in Mbuji Mayi had the strongest preferences for purified water. The highest and lowest WTPs were found in only two towns: households in Tshishimbi had the highest WTPs for reliable or purified water, despite their lower preferences for both compared to other towns; and households in Tshitenge had the lowest WTPs for both, despite their higher preferences for both compared to other towns. These disparities in preference and WTP highlight the complexities of purchase decisions and valuation in consumers. Also, these results represent stated preferences and WTPs, which are typically less indicative of actual purchase decisions than revealed preferences and WTPs [38]; additional research of actual purchases (e.g., sales data from WSPs) would be required to evaluate revealed preferences and WTPs for reliable or purified water. All reported results were statistically significant with p = 0.000 except for WTP in Tshishimbi, where p = 0.02.

## Perceptions, norms, and abilities

In regards to household norms, almost all (94%) of households reported no treatment at the household level. The primary collection individual was female at 91% of households. For

**Table 2. Number (%) sanitary inspection issues identified during collection point surveys, n = 70.**

| Sanitary inspection | n (%) |
|---|---|
| Heavy vegetation | 23 (33) |
| Leaking tap | 54 (77) |
| Unclean tap attachments | 52 (74) |
| Inadequate drainage | 68 (97) |
| Inadequate fencing/barrier | 69 (99) |
| Signs of nearby pollution | 24 (34) |

**Table 3. Preference and willingness to pay for reliable water and purified water with errors.**

| WSP Area | Reliable | Purified | Number of Households |
|---|---|---|---|
| SADEL | 1.55±0.08 | 1.03±0.09 | 239 |
| | 551±82 CDF | 364±60 CDF | |
| ACAEL | 0.95±0.09 | 1.54±0.11 | 218 |
| | 557±155 CDF | 902±243 CDF | |
| FOMI Mbuji Mayi | 1.48±0.16 | 1.88±0.17 | 100 |
| | 466±112 CDF | 592±137 CDF | |
| CAEPT | 1.42±0.16 | 1.61±0.16 | 100 |
| | 868±382 CDF | 985±430 CDF | |
| FOMI Tshitenge | 1.50±0.10 | 1.39±0.11 | 173 |
| | 425±63 CDF | 393±61 CDF | |
| All Areas | 1.37±0.05 | 1.39±0.05 | 830 |
| | 532±49 CDF | 540±50 CDF | |

sanitation facilities, 90% of respondents used varying types of pit latrines. The majority (62%) of respondents share a toilet facility, with the median number of users of toilet facilities being 12 people. About 12% of households reported having a handwashing facility, and enumerators' observations confirmed this was roughly accurate.

The survey asks individuals questions to assess their perceptions of using clean water, such as risk or importance of clean water, confidence in availability, and planning for clean water collection. Households expressed a high perceived risk of diarrheal disease with 88% of respondents indicating that their child under five years of age had a "high" or "very high" risk of contracting diarrhea. The vast majority of respondents correctly identified causes of diarrhea as contaminated food and water (88% and 90% respectively), and this was fairly consistent across areas. Overall, two-thirds of respondents stated getting clean water is "time consuming" or "very time consuming," in line with previous results indicating about half of households spend over an hour collecting water. When asked about their perceptions of others' behavior, the majority (65%) of respondents said that "almost nobody" or "less than half" of the people in their community use clean water. However, 76% of households stated that it was either "important" or "very important" for people they respect or are important to them to drink clean water.

## Health metrics

For children under five, households were asked if they had experienced certain health issues in the past seven days. Respondents reported about 26% of children under five in their household had experienced diarrhea in the last week, with 26% experiencing watery stool and 38% experiencing three or more bowel movements in a single day.

Of the 1,275 individuals for whom a date of birth was reported and age in months was calculated, 1,030 of those records were for children in the 3–60-month age range for which WHO standards for MUAC by age are available and household ID information was present. (Table 4).

## Water quality assessments

Water quality assessments are conducted at both household points-of-consumption and water service provider points-of-collection. These temporal household water quality assessments target several parameters including turbidity, pH, conductivity, free and total chlorine, nitrate, nitrite, manganese, arsenic, fluoride, and thermotolerant coliforms. Water quality assessments

**Table 4. Descriptive statistics for MUAC measurements compared to WHO standards for children aged 3–60 months, stratified by age (in years) and sex..**

| Age | Sex | N | Present | MUAC | Mean (SD) |
|---|---|---|---|---|---|
| 0 | Female | 39 | 13 | 13 | 14.7 (1.4) |
| 0 | Male | 23 | 4 | 4 | 16.0 (2.8) |
| 1 | Female | 97 | 58 | 58 | 14.3 (1.6) |
| 1 | Male | 110 | 62 | 62 | 14.7 (1.6) |
| 2 | Female | 154 | 115 | 115 | 14.7 (1.8) |
| 2 | Male | 125 | 90 | 90 | 15.0 (1.5) |
| 3 | Female | 110 | 79 | 78 | 15.5 (1.4) |
| 3 | Male | 114 | 82 | 82 | 15.5 (2.0) |
| 4 | Female | 115 | 79 | 79 | 15.8 (1.6) |
| 4 | Male | 121 | 73 | 73 | 15.3 (1.6) |
| 5 | Female | 12 | 4 | 4 | 15.5 (1.0) |
| 5 | Male | 10 | 6 | 6 | 15.5 (1.4) |

help locate possible areas of contamination and inform possible interventions on treatment and storage practices. Table 5 displays the source type at each WSP and number of collection points.

Water quality surveying was launched in households within 500m of ACAEL, and initial coliform counts and nitrate values exceeded equipment thresholds. Based upon the pilot results, the team pivoted methodology and subsequently the results of thermotolerant coliform presented in Table 6 include baseline data at all WSP functional collection points except ACAEL, where the second round water quality testing is presented.

Point-of-consumption data at sampled households within 500m of each WSP is presented in Table 7. Almost all WSP collection points (100%) and household points-of-consumption (98%) had presence of thermotolerant coliforms. Additionally, almost all (99%) water samples did not show presence of free or total chlorine, indicating the lack of or insufficient chlorination. Turbidity was an issue at all WSPs except for FOMI Mbuji Mayi. At the time of sampling, 44% of households had collected their drinking water from the nearby WSP. The greatest WSP usage was seen at FOMI Mbuji Mayi (100%) and CAEPT (88%).

There was no arsenic presence or fluoride concerns at any WSPs, but only 17% of respondents had a collection time under 30 minutes round trip, and only 2% of households did not have thermotolerant coliforms in their drinking water. Therefore, virtually no households currently have safely managed drinking water services.

## Conclusion

This study builds on existing research in various ways. This research adds to the body of evidence about effectiveness of various WASH interventions in differing contexts. Our design

**Table 5. WSPs with water quality assessments conducted, with source type and collection points identified.**

| WSP Area | Source Type | Water Points |
|---|---|---|
| SADEL | Borehole | 45 |
| ACAEL | Spring | 6 |
| FOMI Mbuji Mayi | Borehole | 21 |
| CAEPT | Borehole | 31 |
| FOMI Tshitenge | Spring | 16 |

**Table 6. Percentage of thermotolerant coliform counts within each World Health Organization risk threshold, with samples from functional WSP collection points..**

| | No Risk (0 CFUs/100mL) n(%) | Low Risk (0–1 CFUs/100mL) n(%) | Intermediate Risk (1–10 CFUs/100mL) n(%) | High Risk (10–100 CFUs/100mL) n(%) | Very High Risk (>100 CFUs/100mL) n(%) |
|---|---|---|---|---|---|
| SADEL n = 14 | 0 (0) | 1 (7) | 1 (7) | 5 (36) | 7 (50) |
| ACAEL n = 6 | 0 (0) | 0 (0) | 3 (50) | 2 (33) | 1 (17) |
| FOMI Mbuji Mayi n = 20 | 0 (0) | 0 (0) | 7 (35) | 6 (30) | 7 (35) |
| CAEPT n = 13 | 0 (0) | 1 (8) | 3 (23) | 6 (46) | 3 (23) |
| FOMI Tshitenge n = 1 | 0 (0) | 0 (0) | 1 (100) | 0 (0) | 0 (0) |

will evaluate interventions ranging from infrastructure to financial capacity to governance, specific to the needs of the WSPs. Through the water service level, water quality, health indicators, and household level metrics this study will assess whether implemented interventions at water service providers were first, effective, and second, resulted in improved water quality and provision for households.

Third, while a randomized trial is not feasible for this study given the logistical constraints, this study combines a robust and flexible study design with state-of-the-art statistical methods in order to estimate the causal impacts of program activities. Our design uses a causal road map approach to identify statistical parameters of interest and appropriate statistical estimators. By studying the likely set of measured and unmeasured confounders, this study will assess the ability to interpret effects causally [49]. This approach combines rigorous summative evaluation of the effectiveness of a given intervention with formative evaluation of the progress and effectiveness of intervention implementation [50]. In real-time and parallel with program deployment, this methodology has the advantage of providing statistically credible while concurrently actionable data.

**Table 7. Thermotolerant coliform counts within each World Health Organization risk threshold, with samples from points-of-consumption at households within 500m of WSP collection points..**

| | No Risk (0 CFUs/100mL) n(%) | Low Risk (0–1 CFUs/100mL) n(%) | Intermediate Risk (1–10 CFUs/100mL) n(%) | High Risk (10–100 CFUs/100mL) n(%) | Very High Risk (>100 CFUs/100mL) n(%) |
|---|---|---|---|---|---|
| SADEL n = 51 | 0 (0) | 0 (0) | 0 (0) | 3 (6) | 48 (94) |
| ACAEL n = 25 | 0 (0) | 0 (0) | 0 (0) | 4 (16) | 21 (84) |
| FOMI Mbuji Mayi n = 25 | 3 (12) | 0 (0) | 1 (4) | 5 (20) | 16 (64) |
| CAEPT n = 25 | 0 (0) | 0 (0) | 0 (0) | 2 (8) | 23 (92) |
| FOMI Tshitenge n = 51 | 0 (0) | 0 (0) | 1 (2) | 6 (12) | 44 (86) |

Ultimately, this implementation science strategy incorporates measurement during program deployment of service performance and feeds these results back to Chemonics, WSPs, and communities. This methodology seeks to enable iterative testing and improvement throughout the life cycle of the program to dynamically improve the quality and sustainability service delivery of WSPs in Kasai Oriental.

## Supporting information

**S1 File.**
(PDF)

## Acknowledgments

We thank all the participants who contributed to this study. We also thank the staff in Mbuji-Mayi for their tireless efforts in conducting the community-level data collection.

## Author Contributions

**Conceptualization:** Kathleen Kirsch, Corey Nagel, Elizabeth Carlton, Amy Javernick-Will, Karl Linden, Evan Thomas.

**Data curation:** Kathleen Kirsch, Corey Nagel, Andrea J. Lund, James Harper.

**Formal analysis:** Kathleen Kirsch, Corey Nagel, Elizabeth Carlton.

**Funding acquisition:** Amy Javernick-Will, Karl Linden, Evan Thomas.

**Investigation:** Kathleen Kirsch, John Ecklu, Ghislaine Akonkwa Zawadi, Pacifique Mugaruka Ntabaza, Christina Barstow, Evan Thomas.

**Methodology:** Corey Nagel, Christina Barstow, Elizabeth Carlton, Amy Javernick-Will, Karl Linden, Evan Thomas.

**Project administration:** Chantal Iribagiza, Ghislaine Akonkwa Zawadi, Pacifique Mugaruka Ntabaza, Christina Barstow, Amy Javernick-Will, Evan Thomas.

**Resources:** Evan Thomas.

**Supervision:** Chantal Iribagiza, Elizabeth Carlton, Amy Javernick-Will, Karl Linden, Evan Thomas.

**Visualization:** Kathleen Kirsch.

**Writing – original draft:** Kathleen Kirsch, Corey Nagel, Christina Barstow, Evan Thomas.

**Writing – review & editing:** Chantal Iribagiza, John Ecklu, Ghislaine Akonkwa Zawadi, Pacifique Mugaruka Ntabaza, Elizabeth Carlton, Amy Javernick-Will, Karl Linden, Evan Thomas.

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
