## [Decision Letter · Decision Letter 0]

6 Oct 2022

PONE-D-22-20003Study design to evaluate water service provision among peri-urban communities in Kasai Oriental and Lomami Provinces, Democratic Republic of the CongoPLOS ONE

Dear Dr. Thomas,

Thank you for submitting your manuscript to PLOS ONE. After careful consideration, we feel that it has merit but does not fully meet PLOS ONE’s publication criteria as it currently stands. Therefore, we invite you to submit a revised version of the manuscript that addresses the points raised during the review process. Reviewers 2 and 3 provide some detailed comments to address which I don't think should pose much of a problem.   Reviewer 1 doesn't get the point of the manuscript but nevertheless has some comments that you could consider.

We look forward to receiving your revised manuscript.

Kind regards,

Alison Parker

Academic Editor

PLOS ONE

Journal Requirements:

3. Please amend your current ethics statement to address the following concerns:

a) Did participants provide their written or verbal informed consent to participate in this study?

Reviewers' comments:

Reviewer's Responses to Questions

**Comments to the Author**

1. Does the manuscript provide a valid rationale for the proposed study, with clearly identified and justified research questions?

Reviewer #1: Partly

Reviewer #2: Partly

Reviewer #3: Partly

2. Is the protocol technically sound and planned in a manner that will lead to a meaningful outcome and allow testing the stated hypotheses?

Reviewer #1: Partly

Reviewer #2: Yes

Reviewer #3: Yes

3. Is the methodology feasible and described in sufficient detail to allow the work to be replicable?

Reviewer #1: Yes

Reviewer #2: Yes

Reviewer #3: Yes

4. Have the authors described where all data underlying the findings will be made available when the study is complete?

Reviewer #1: Yes

Reviewer #2: Yes

Reviewer #3: Yes

5. Is the manuscript presented in an intelligible fashion and written in standard English?

Reviewer #1: Yes

Reviewer #2: Yes

Reviewer #3: Yes

6. Review Comments to the Author

You may also provide optional suggestions and comments to authors that they might find helpful in planning their study.

Reviewer #1: This paper is not recommended for publication for the following reasons:

1. It presents a study design rather than a completed study. The novel contributions to knowledge are limited, since most of the presented study design draws on established methodologies, with few new methods.

2. There is no information about the budget available to conduct this study nor the personnel available. It is difficult to judge therefore how applicable this study design would be to other settings.

3. There is no information about the ethical considerations inherent in the study and how these will be addressed. For example, if the health assessment identifies poor health in the study subjects, will they be offered treatment? Same for other aspects of the study (e.g. water quality).

4. The paper should have discussed the possible downsides/biases of their approaches and which alternative approaches were considered and dismissed. That would have been a more useful contribution to the literature on this subject.

Reviewer #2: This manuscript describes a well-designed protocol for evaluating the impact of water service provision inteventions in DRC using an implementation science framework. The proposed study design and analytical methods are sound and will be an important contribution to the evidence base for WASH intervention effectiveness.

Major comments

• The manuscript would benefit from a clear and early (e.g. in Background section) description of the research question, including 1) the intervention activities, 2) outcomes being evaluated.

• The manuscript presents preliminary data, but this is not mentioned in the title, abstract, or background section. I suggest re-framing this paper as “study design and preliminary baseline results”

Minor comments:

• Verb tense changes throughout, particularly in the design section. Since these decisions presumably have already been made (and likely implemented), use of past tense is suggested.

• Suggest including line numbers to facilitate next round of review

• Make sure all acronyms are defined at first use

• The preliminary results often repeat methods earlier described; methods should not be included in results section.

Specific comments

Abstract:

• “…to establish the impact of interventions on peri-urban water service provision” – please define specifically your outcomes of interest (water provision is pretty broad)

• Briefly include information on sampling and data collection methods – for example household-based measures are referenced (water security, health status), but how many households will be surveyed, and how will they be selected?

• Final sentence: What are the program outcomes and how will success be defined?

Background

• Define WASH at first use

• Paragraph 6 (about ITS) would fit better in the methodology section

• The intervention and research question is unclear. Please state what the specific intervention(s) are and what specific outcomes are being evaluated.

Design

Study setting

• How were the 32 WSPs selected?

Study design

• How were WSPs included in study randomly selected?

o Edited to add: this is then described in the subsequent paragraph. This section is therefore repetitive and should be made more concise. No need to state in the first paragraph that you will randomly select WSPs and in the next paragraph that you did select WSPs. Please complete a through review of the methods for conciseness.

• Who is going to be targeted for the household survey, and how are they selected? What happens if they aren’t there for a visit? Can someone else be surveyed?

Sample size

• How are you defining diarrhea?

Study Rationale

• This is exactly what I was hoping to see in the background section. I suggest moving all but the paragraph starting with “Figure 3” to background so the audience knows what the interventions include and what outcomes you are evaluating

Eligibility

• There is no eligibility criteria described here, other than being within 500 meters of a standpipe operated by the WSPs. Is there other eligibility criteria?

• Suggest moving this up to the study design section

Study components

• Suggest clarifying that these surveys are conducted among the selected households.

• Surveys “on 12 water service providers” wasn’t previously described – who are you surveying within these providers?

• Please state whether poverty scorecard tool is validated (+ citation, if available)

• WTP evaluation – (last sentence) – will the DCE evaluation be conducted among the selected households or elsewhere? This is unclear based on current phrasing.

• Health – how is diarrhea defined

o What do you mean by “complemented by public health records”? Do you mean verify with medical records for illness events?

• Water quality assessments – 25 households per WSP in the study, correct? Not 25 households total. Please clarify and also include overall total of households undergoing water quality assessments.

• QCA – in the last paragraph in this section, the sentence starting with “we will then assess QCA metrics…” is very hard to follow.

Statistical methods

• All models will be adjusted for both individual and WSP-level characteristics in order to minimize

potential selection bias and confounding resulting from the non-random timing of program delivery – suggest giving examples of these characteristics and how they will be determined

• 2nd paragraph says “section ?”

Preliminary data

• Suggest including in abstract, background (and title) that this paper will also present preliminary baseline data, since this hasn’t been mentioned until this point

• Were the households included in the baseline ultimately enrolled in the study? Unclear if this is a part of the larger study or just preliminary/exploratory work

• How many households were included in the baseline survey? Is it 669 (as specified in the demographics and poverty scorecard section?) if so, please move up as it is unclear whether 669 is how many surveys were administered or how many completed the scorecard.

• How is “primary water source” defined?

• Please also present water sources as categorized improved/unimproved since that was what was described in the methods

• Willingness to pay – consider conversion to USD or present conversion rate

• MUAC – the first part of this paragraph (defining MUAC and how it’s measured/used) belongs in the methods.

• Water quality assessments – how many households total?

• Water quality assessments – the final paragraph (describing JMP criteria) belongs in methods

Discussion

• Last sentence of first paragraph contains typos

• The estimation strategies belongs in methods – do not introduce new ideas in discussion section

Reviewer #3: The manuscript presents the study protocol for an evaluation of a USAID intervention designed to improve water service provision in peri-urban communities in two provinces in the Democratic Republic of the Congo. The study aims to overcome challenges inherent with some monitoring practices that fail to assess the overall project performance. It pilots an implementation science impact evaluation approach to attempt to overcome the issues associated with traditional impact evaluations (e.g. the lengthy processes). The study is designed to assess performance against multiple indicators and will use a range of data collection methods. The study is timely, as it identifies the tension that exists in monitoring and evaluation (M&E) between prompt feedback that can contribute directly to project activities, and rigour.

The manuscript could be strengthened by developing the opening paragraphs of the Background section. Stronger engagement with previously published research would enhance the section and provide clearer context. It appears that there is a strong rationale for research into how best to balance M&E rigour with prompt results but refining and tightening the Background section would help this message to come to the fore. Relatedly, further information on the intervention that the study is designed to evaluate would be useful for context. While more information on the intervention as a whole would be beneficial, specific details on whether all the water service providers in the area will receive an intervention would be helpful, as would clarity on the timing of the intervention in relation to the data collection. Details of whether any other interventions are taking place or scheduled to take place that could impact the study would also be useful.

On a related point, clarification on whether information will be obtained to identify changes in the water point that participants use, factors driving those changes and whether/how changes in water point could impact the inferences of the study would be beneficial. The wording of the Study components section suggests that the methods are not comprehensively outlined in the protocol. The description needs to provide adequate details for the protocol to be reproduced and replicated. Further information may be required to achieve this. In Figure 3, for example, it looks like remote sensing climate/weather data will be measured but this is not elaborated on in the text.

The manuscript would benefit from further proofreading. For example, the acronym WASH and the initialism ETD should be spelt out in full in the first instance. In Section 4 (statistical methods), on the third line of the second paragraph, authors are signposted to a section, but the specific section is marked by a question mark. In the references list, missing information in reference 34 is marked by question marks and reference 45 looks to be incomplete, with the URL in the main body of the text, but limited information in the references list.

On a minor point, Figure 2 could perhaps be reformatted to improve clarity. Deleting the empty rows, for example, may make the schedule look cleaner.

Clarification on the form of consent obtained (written/oral) and when it will be obtained would also strengthen the manuscript. There appears to be limited information on the data management plan, so further details on this would be useful.

7. PLOS authors have the option to publish the peer review history of their article (what does this mean?). If published, this will include your full peer review and any attached files.

Reviewer #1: No

Reviewer #2: No

Reviewer #3: No

---

## [Decision Letter · Decision Letter 1]

3 Feb 2023

PONE-D-22-20003R1Study design and baseline to evaluate water service provision among peri-urban communities in Kasai Oriental, Democratic Republic of the CongoPLOS ONE

Dear Dr. Thomas,

Thank you for submitting your manuscript to PLOS ONE. After careful consideration, we feel that it has merit but does not fully meet PLOS ONE’s publication criteria as it currently stands. Therefore, we invite you to submit a revised version of the manuscript that addresses the points raised during the review process. Many thanks for addressing the comments. There are just a few minor comments left to address from both reviewers.

We look forward to receiving your revised manuscript.

Kind regards,

Alison Parker

Academic Editor

PLOS ONE

Journal Requirements:

Reviewers' comments:

Reviewer's Responses to Questions

**Comments to the Author**

1. Does the manuscript provide a valid rationale for the proposed study, with clearly identified and justified research questions?

Reviewer #2: Yes

Reviewer #3: Yes

2. Is the protocol technically sound and planned in a manner that will lead to a meaningful outcome and allow testing the stated hypotheses?

Reviewer #2: Yes

Reviewer #3: Yes

3. Is the methodology feasible and described in sufficient detail to allow the work to be replicable?

Reviewer #2: Yes

Reviewer #3: Yes

4. Have the authors described where all data underlying the findings will be made available when the study is complete?

Reviewer #2: Yes

Reviewer #3: No

5. Is the manuscript presented in an intelligible fashion and written in standard English?

Reviewer #2: Yes

Reviewer #3: Yes

6. Review Comments to the Author

You may also provide optional suggestions and comments to authors that they might find helpful in planning their study.

Reviewer #2: A few minor comments:

• First sentence in abstract - recommend adding “results” after “We present study design and baseline”

• Lines 345-349 remains very difficult to follow. Suggest breaking into sentences for each component of the QCA metrics.

• Line 423 – suggest replacing the word “significant” with another adjective because of connotation with “statistically significant”, particularly since this is the results section

• Line 544 – this should be effectiveness not efficacy

Reviewer #3: Thank you for taking the time to address many of the comments. The manuscript could be enhanced further by making additional edits to the Introduction. For example, lines 71 - 75 are very similar to lines 88 - 94 so there is some slight repetition there. Addressing this, and restructuring the Introduction section more broadly could improve the flow and focus of the initial part of the manuscript. Also, a further check through the in-text citations to ensure that all relevant information is referenced, would be beneficial. It is unclear, for example, where the information provided in lines 46/47 was obtained from.

7. PLOS authors have the option to publish the peer review history of their article (what does this mean?). If published, this will include your full peer review and any attached files.

Reviewer #2: No

Reviewer #3: No

---

## [Editor Report · Decision Letter 2]

9 Feb 2023

PONE-D-22-20003R2Study design and baseline to evaluate water service provision among peri-urban communities in Kasai Oriental, Democratic Republic of the CongoPLOS ONE

Dear Dr. Thomas,

Thank you for submitting your manuscript to PLOS ONE. After careful consideration, we feel that it has merit but does not fully meet PLOS ONE’s publication criteria as it currently stands. Therefore, we invite you to submit a revised version of the manuscript that addresses the points raised during the review process. I cannot see that any of the changes requested by the reviewer have been made, despite what is claimed in the response document, have been made and I wonder if the wrong version of the manuscript has been resubmitted in error.

We look forward to receiving your revised manuscript.

Kind regards,

Alison Parker

Academic Editor

PLOS ONE
---

## [Author Response · Author response to Decision Letter 2]

9 Feb 2023

See enclosed (as previously provided)

---

## [Editor Report · Decision Letter 3]

14 Feb 2023

PONE-D-22-20003R3Study design and baseline to evaluate water service provision among peri-urban communities in Kasai Oriental, Democratic Republic of the CongoPLOS ONE

Dear Dr. Thomas,

Thank you for submitting your manuscript to PLOS ONE. After careful consideration, we feel that it has merit but does not fully meet PLOS ONE’s publication criteria as it currently stands. Therefore, we invite you to submit a revised version of the manuscript that addresses the points raised during the review process.

We look forward to receiving your revised manuscript.

Kind regards,

Alison Parker

Academic Editor

PLOS ONE

Journal Requirements:

Additional Editor Comments:

The same version has been submitted, I can tell from the date stamp (Feb 6th).   The problem is that the changes have not been made in this version.   Please submit a revised version wioth the minor changes made.

---

## [Editor Report · Decision Letter 4]

17 Feb 2023

PONE-D-22-20003R4Study design and baseline to evaluate water service provision among peri-urban communities in Kasai Oriental, Democratic Republic of the CongoPLOS ONE

Dear Dr. Thomas,

Thank you for submitting your manuscript to PLOS ONE. After careful consideration, we feel that it has merit but does not fully meet PLOS ONE’s publication criteria as it currently stands. Therefore, we invite you to submit a revised version of the manuscript that addresses the points raised during the review process. You've now completed the first correction but all others remain outstanding.   Please complete all corrections and ensure the correct version of the manuscript is uploaded.

We look forward to receiving your revised manuscript.

Kind regards,

Alison Parker

Academic Editor

PLOS ONE
---

## [Editor Report · Decision Letter 5]

1 Mar 2023

Study design and baseline to evaluate water service provision among peri-urban communities in Kasai Oriental, Democratic Republic of the Congo

PONE-D-22-20003R5

Dear Dr. Thomas,

We’re pleased to inform you that your manuscript has been judged scientifically suitable for publication and will be formally accepted for publication once it meets all outstanding technical requirements.

Kind regards,

Alison Parker

Academic Editor

PLOS ONE
---

## [Editor Report · Acceptance letter]

4 Apr 2023

PONE-D-22-20003R5 

Study design and baseline to evaluate water service provision among peri-urban communities in Kasai Oriental, Democratic Republic of the Congo 

Dear Dr. Thomas:

I'm pleased to inform you that your manuscript has been deemed suitable for publication in PLOS ONE. Congratulations! Your manuscript is now with our production department. 

Kind regards, 

on behalf of

Dr. Alison Parker 

Academic Editor

PLOS ONE